# Reversed size-dependent stabilization of ordered nanophases

J. Pirart[1], A. Front[2], D. Rapetti [3], C. Andreazza-Vignolle[1], P. Andreazza [1], C. Mottet[2] & R. Ferrando [3]

The size increase of a nanoscale material is commonly associated with the increased stability of its ordered phases. Here we give a counterexample to this trend by considering the formation of the defect-free $L1_1$ ordered phase in AgPt nanoparticles, and showing that it is better stabilized in small nanoparticles (up to 2.5 nm) than in larger ones, in which the ordered phase breaks in multiple domains or is interrupted by faults. The driving force for the $L1_1$ phase formation in small nanoparticles is the segregation of a monolayer silver shell (an Ag-skin) which prevents the element with higher surface energy (Pt) from occupying surface sites. With increasing particle size, the Ag-skin causes internal stress in the $L1_1$ domains which cannot thus exceed the critical size of ~2.5 nm. A multiscale modelling approach using full-DFT global optimization calculations and atomistic modelling is used to interpret the findings.

---

[1] Université d'Orléans, CNRS, ICMN UMR7374, 1b rue de la Férollerie, 45071 Orléans Cedex 2, France. [2] Aix-Marseille Université, CNRS, CINaM UMR 7325, Campus de Luminy, 13288 Marseille, France. [3] Physics Department, University of Genoa, Via Dodecaneso 33, 16146 Genoa, Italy. Correspondence and requests for materials should be addressed to C.A.-V. (email: caroline.andreazza@univ-orleans.fr) or to P.A. (email: pascal.andreazza@univ-orleans.fr) or to C.M. (email: mottet@cinam.univ-mrs.fr) or to R.F. (email: ferrando@fisica.unige.it)

Increasing the size of a nanoscale material is commonly associated to a widening of the stability domain of its ordered phases at equilibrium[1]. Well-known examples are the melting point temperature increase with nanoparticle size[2], which widens the stability range of the ordered solid phase, and the increase of the order−disorder transition temperature in nanoalloys, such as CoPt, to approach the bulk limit[3,4]. Here we show that this is not a general rule, since an increase of the system size can have the opposite effect: in small AgPt clusters, up to $D \sim 2.5$ nm, a well-ordered intermetallic phase is stabilized at equilibrium, but, above this size, either the ordered phase breaks down into less-ordered multidomains, or its regular arrangement is interrupted by faults. The small-size stable nanophase is characterized by a specific surface segregation effect, while stress accumulation contributes to the breaking of the ordered phase in larger objects.

The bulk AgPt system shows a very atypical phase diagram in the Ag-based family, with a large miscibility gap between $Ag_{88}Pt_{12}$at% and $Ag_2Pt_{98}$at% at low temperature[5,6] and an ordered phase observed only for an extremely narrow composition range around $Ag_{50}Pt_{50}$at%[7]. This phase was observed by HAADF-STEM on bulk samples as made of small $L1_1$ ordered patches of few nanometers, irregularly alternating with disordered regions[7]. The $L1_1$ phase is quite difficult to obtain experimentally. Remarkably, an intermetallic compound has been obtained by annealing to 700 °C AgPt randomly intermixed nanoparticles[8] (NPs). This compound was not an $L1_1$ intermetallic, but a new material with hexagonal and cubic close-packed sequences of alternating Ag and Pt planes. As regards AgPt nanoparticles, alloyed and core@shell Pt@Ag structures have been produced[9–14], but without indications of $L1_1$ phases up to now. As we show in the following, the $L1_1$ phase can indeed be obtained in AgPt nanoparticles, as the result of a subtle interplay of different physical effects.

AgPt nanoparticles find application in catalysis and in fuel cells[15–17], with the optimal chemical ordering depending on the reaction. In any case, a crucial point is the catalytic efficiency in terms of durability. This implies that the problem of structural evolution of the nanoparticles must be addressed when dealing with applications. This evolution may be either due to a non-equilibrium state of the as-synthesized structure or to the operando conditions. Therefore, assessing the thermodynamic stability of AgPt nanoparticles is very important for applications.

In the following we experimentally show that nanoparticles with $L1_1$ ordered domains are obtained for sizes up to ~2.5 nm.

We demonstrate that in these nanoparticles the $L1_1$ domains are surrounded by an extremely thin (mostly monoatomic) Ag surface layer. This structure is denoted as $L1_1$@Ag-skin structure. For larger sizes, the ordered phase breaks into multiple $L1_1$ domains, or it is interrupted by faults. These results are rationalized by a combination of density functional theory (DFT) global optimization searches and of atomistic modelling. The calculations show that the $L1_1$ phase is indeed stabilized by Ag surface segregation in small nanoparticles, and that the breaking of the $L1_1$ phase in larger nanoparticles is due to the accumulation of stress in the structures.

## Results

**$L1_1$ phase formation.** Ag and Pt atoms were codeposited at room temperature on the amorphous carbon grid, to achieve an average composition close to the equi-composition. For $2.3 \times 10^{15}$ at $cm^{-2}$, transmission electron microscopy (TEM) images show an assembly of isolated NPs with mean diameter $D = 1.5$ nm (see Supplementary Fig. 1). For $3.2 \times 10^{15}$ at $cm^{-2}$, static coalescence occurs due to the NPs close vicinity. Interdiffusion bridges form between neighbour particles leading to ramified objects, composed of 2–4 elementary particles with $D = 1.9$ nm on average. Grazing incidence small angle X-ray scattering (GISAXS) measurements confirm the average $D$ and gives $H/D = 0.76$ ($H$ is the average height, see Supplementary Fig. 2). $H/D$ is lower than 1 because supported NPs look like truncated spheres due to the support-metal interaction. This value was taken into account for defining models used for simulated STEM-HAADF images. HRTEM images show an fcc structure with some incomplete surface planes with respect to the perfect truncated octahedron (TOh) morphology. Particles present many defects such as twins and stacking faults (Fig. 1a, b). On the fast Fourier transform (FFT, see the inset of Fig. 1a) of a large number of particles, $d$-spacings of $0.232 \pm 0.005$ and $0.201 \pm 0.005$ nm are obtained. They correspond to (111) and (200) interplanar distances of an fcc structure with lattice parameter $a = 0.402 \pm 0.005$ nm, closer to the parameter of Ag ($a = 0.4086$ nm) than to that of Pt ($a = 0.392$ nm). A low magnification STEM-HAADF image of the as-prepared sample (Fig. 1c) shows NPs with homogeneous contrast. Figure 1d shows a high-resolution HAADF-STEM image along the [01$\bar{1}$] zone axis and its corresponding FFT. The lack of superstructure reflections in the FFT indicates that there is no order in the [111] direction. Simulated HAADF images of nanoparticles exhibiting a completely random solid solution and

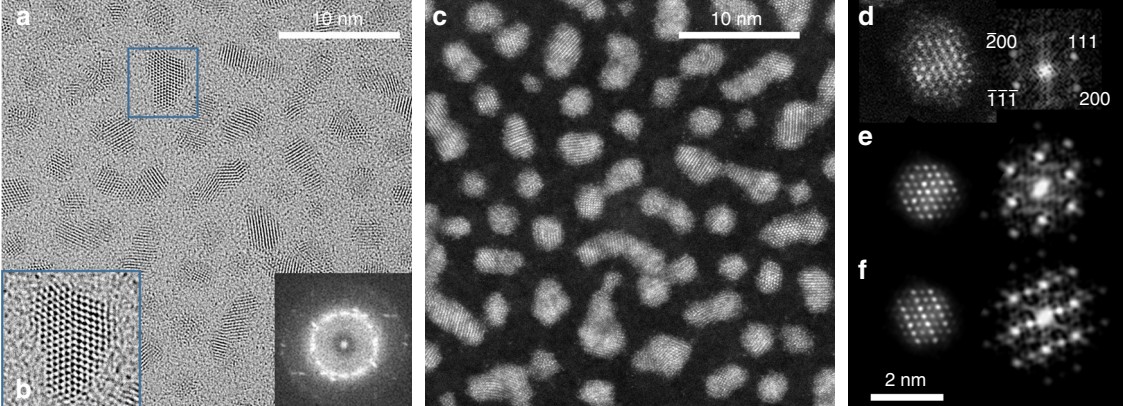

**Fig. 1** As-grown room temperature AgPt sample (flux $3.2 \times 10^{15}$ at $cm^{-2}$). **a** HRTEM images and in the inset, the total Fourier transform. **b** Zoom (×2) showing crystal twinning. **c** STEM-HAADF image with homogeneous contrast. High-resolution STEM-HAADF image and the corresponding FFT for **d** experimental NP along the [01$\bar{1}$] zone axis and the corresponding simulated images **e** for a random solid solution and **f** for an $L1_1$ ordered structure. HRTEM high resolution in TEM, STEM-HAADF high angle annular dark field in scanning transmission electron microscope, NP nanoparticle

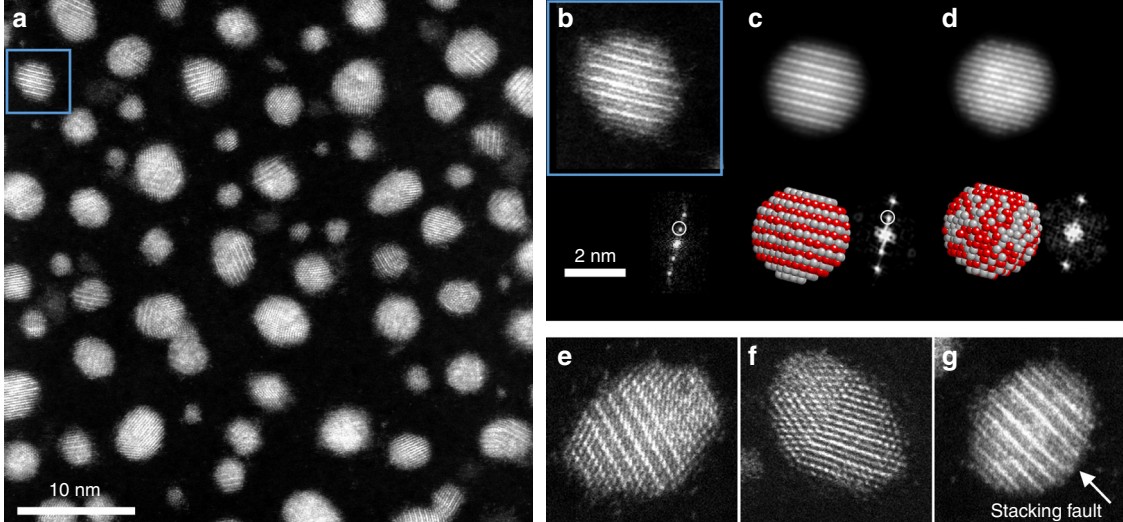

**Fig. 2** AgPt NPs after annealing at 400 °C. **a** STEM-HAADF image; **b** a zoom on one NP oriented along the [01$\bar{1}$] zone axis showing the L1$_1$ phase with alternating planes of Ag and Pt and its corresponding FFT showing the superstructure reflections. **c**, **d** Simulated STEM-HAADF images along the [01$\bar{1}$] zone axis for an L1$_1$ structure and a random solid solution respectively. **e**–**g** show multidomains, twin and a stacking fault for L1$_1$ NPs (Pt atoms are in red and Ag atoms in grey). The scale is the same for **b**–**g**. NP nanoparticle, FFT fast Fourier Ttransform, STEM-HAADF high angle annular dark field in scanning transmission electron microscope

an L1$_1$ ordered structure are shown in Fig. 1e, f, respectively. The experimental image does not match the ordered phase but matches the random solid solution without Ag segregation (A1 phase). The absence of the ordered L1$_1$ phase may be due to slow diffusion at room temperature which prevents equilibration of chemical ordering, causing kinetic trapping into metastable disordered states. This is indeed the case, as demonstrated by our annealing experiments at 400 °C. For the $3.2 \times 10^{15}$ at cm$^{-2}$ sample, TEM images and GISAXS show that the NP shapes and sizes changed. $D$ and $H/D$ increase from 1.9 to 3.0 nm and from 0.76 to 0.82, respectively (see Supplementary Fig. 1 and Supplementary Table 1). This evolution is due to thermally activated diffusion and coalescence, leading to reorganization towards a morphology close to the quasi-spherical equilibrium shape of nanoparticles in vacuum[4,18]. The increase of H/D is indicative of some weakening of the nanoparticle/substrate interaction upon annealing. Atomic-resolution HRTEM images exhibit NPs with TOh structures but with stacking faults and twin boundaries (Supplementary Fig. 1). More important, and contrary to the as-prepared samples, STEM-HAADF images of the annealed samples exhibit particles with non-homogeneous contrast (Fig. 2a). Particles oriented along the [01$\bar{1}$] zone axis or equivalents show strong and weak bright contrasts appearing periodically along the [111] direction (Fig. 2b). On the corresponding FFT pattern, a superstructure reflection corresponding to the half order of the 111 reflection is observed. STEM-HAADF simulated images along the [01$\bar{1}$] zone axis of fully ordered L1$_1$ and random solid solution NPs are shown in Fig. 2c, d. The similarity between observed and simulated images confirms the presence of the L1$_1$ structure, with strong and weak bright contrasts corresponding to Pt and Ag (111) planes, respectively. The distance $d_{111} = 0.230 \pm 0.005$ nm between the consecutive (111) planes corresponds to an fcc lattice parameter $a = 0.400 \pm 0.005$ nm.

To investigate the occurrence of the L1$_1$ phase in the annealed NPs and see whether this occurrence is size-dependent, we analysed a population of 500 NPs. We note that observing the L1$_1$ phase is possible only when the NP is oriented with a zone axis belonging to (111) planes. Assuming that the NPs are randomly oriented on the substrate, one finds that 13% of the NPs have the right orientation. The approximation of random orientations is

crude, but it is useful for comparison with the observations, which are summarized as follows:

The L1$_1$ phase is observed in 24% of the particles (54 occurrences on 230 NPs with planes resolution), i.e. on the whole or on a part of the NP.
The smallest particles ($D < 3$ nm) are mostly monodomain while 85% of large particles ($D > 3$ nm) are multidomain (128 occurrences in 150 NPs), i.e. with antiphase boundaries or twins (Fig. 2e, f). The L1$_1$-phase domain size is between 2 and 3 nm.
Large monodomain particles showing the L1$_1$ phase (6 occurrences on 22 NPs) always exhibit at least one stacking fault corresponding generally to two consecutive pure Ag planes within the NP (cf. Fig. 2g).

Therefore, there is a clear size dependence, since the perfect alternation between pure Pt and Ag (111) planes is only observed in small particles or in small domains (<3 nm) inside larger particles. In these cases, the percentages exceed 13%, indicating that probably a large part of these NPs presents indeed the L1$_1$ phase.

**Surface composition of the Ag-skin**. Since several studies report Ag surface segregation leading to Pt@Ag structures, we characterize the composition of the surface layer of our monodomain L1$_1$ NPs. The analysis of surface contrasts of a 2.8 nm NP is shown in Fig. 3. In order to have both the presence of {111} and {100} facets and the possibility to image the [111] direction, an NP along the [01$\bar{1}$] zone axis was analysed. The experimental image (Fig. 3a–d) was compared to simulated images for a 2.8 nm NP with an aspect ratio of 0.8 according to the GISAXS data. Three structural models were constructed from a fully ordered L1$_1$ domain: single-domain L1$_1$, beginning and finishing with pure Ag(111) or Pt(111) planes (model 1, Fig. 3e–h) and model 2, Fig. 3i–l, respectively, and L1$_1$@Ag-skin, with an Ag layer (mostly a monolayer) covering the L1$_1$ domain (model 3, Fig. 3m–p).

Intensity profiles parallel to (100) and (1$\bar{1}$1) planes were considered. In these profiles, the global intensity variation from the centre to the surface is due to the reduction of the particle thickness in the peripheral region. The intensity variation of two consecutive peaks is attributed to the composition variation of

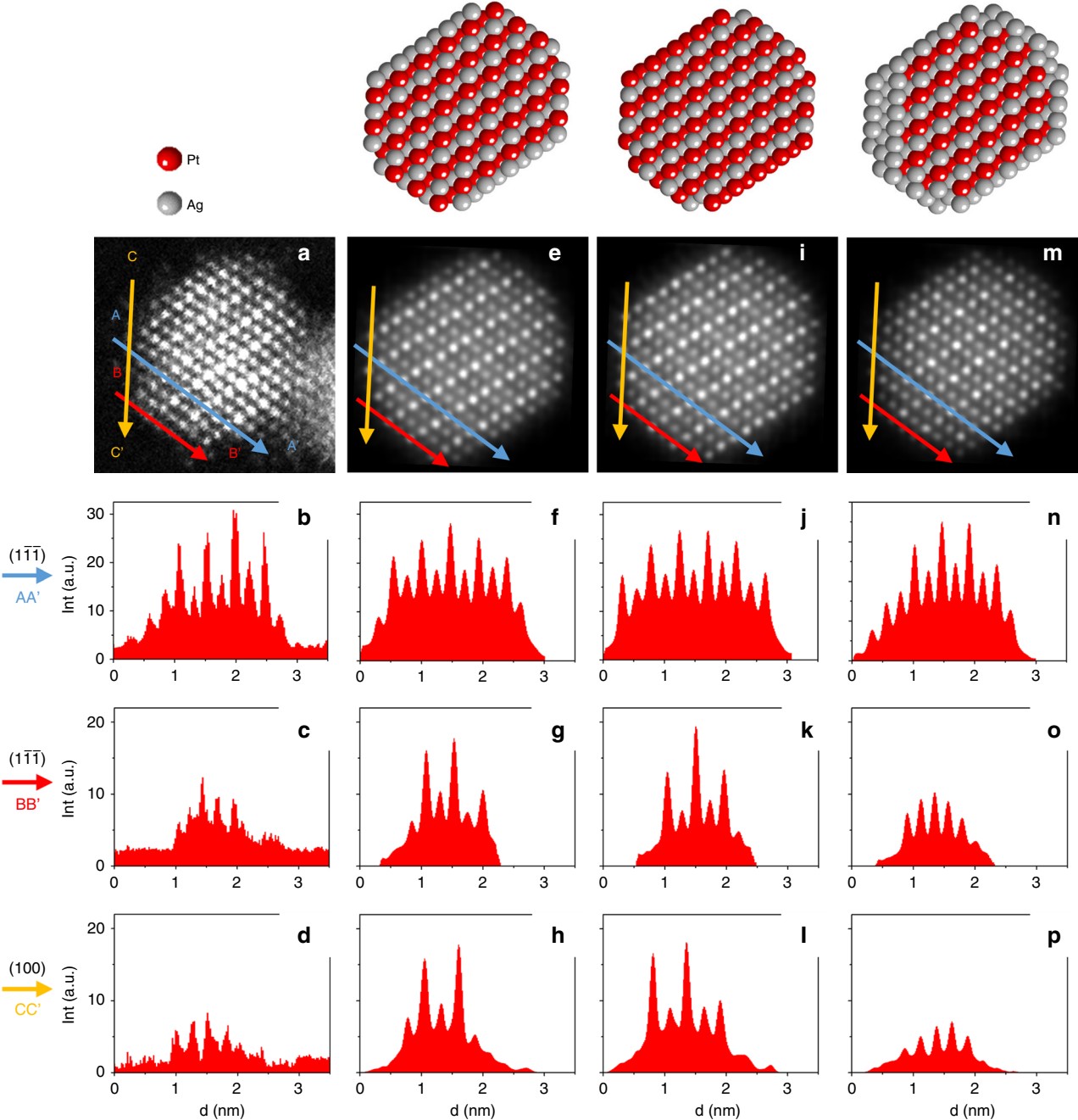

**Fig. 3** L1$_1$@Ag-skin nanoparticle. **a–d** High-resolution STEM-HAADF image of a 2.8 nm NP in the [01$\bar{1}$] zone axis with its corresponding intensity profile lines. **e–h**, **i–l** and **m–p** show the model 1, 2, and 3 in the [01$\bar{1}$] zone axis, respectively, with their corresponding intensity profile lines. The intensity profiles were done for three different locations: along the (1$\bar{1}\bar{1}$) plane close to the centre of the NP (AA'), along the (1$\bar{1}\bar{1}$) plane at the NP surface (BB') and along the (100) plane at the NP surface (CC'). STEM-HAADF high angle annular dark field in scanning transmission electron microscope, NP nanoparticle

[01$\bar{1}$] columns. Close to the NP centre and parallel to the (1 $\bar{1}$ $\bar{1}$) planes, the intensity variation (Fig. 3b, AA' profile line) shows the alternation between bright Pt and darker Ag planes. At both line extremities (A and A'), the intensities correspond to a minimum which can be explained if the atomic columns at the NP surface are pure Ag. Comparison with the simulated profiles indicates that model 2 can be excluded (Fig. 3b, j). To discriminate between models 1 and 3, surface profile lines on the {1$\bar{1}\bar{1}$} and {100} facets (B to B' and C to C' lines, respectively) were collected. Model 1 shows alternating contrasts for both facets (Fig. 3g, h) contrary to model 3 (Fig. 3o, p) and to the experimental profiles (Fig. 3c, d), indicating that (1$\bar{1}\bar{1}$) and (100) surface planes are pure Ag.

Moreover, to fully match the experimental profiles for this 2.8 nm NP, the model is composed of an L1$_1$ core with one monolayer Ag shell except on a {100} facet where there are three consecutive Ag planes (see model 3, Fig. 3).

## Discussion

To rationalize the experimental results we adopt a multiscale modelling approach comprising DFT calculations for small sizes and atomistic modelling for larger sizes.

We checked the stability of the L1$_1$@Ag-skin phase by full-DFT global optimization of chemical ordering in the smallest nanoalloy supporting this phase, a TOh of 79 atoms (Fig. 4a). The perfect

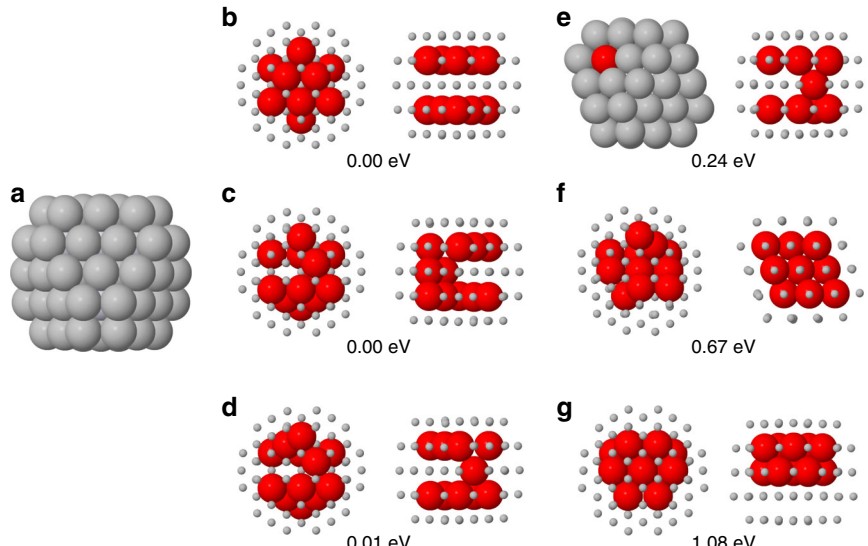

**Fig. 4** DFT results. **a** The truncated octahedron of 79 atoms. This structure has 19 internal sites, i.e. 18 subsurface sites plus the central site. **b–g** Structures of $Ag_{67}Pt_{12}$. Each cluster is shown in two views. Pt atoms are in red and Ag atoms in grey. Pt and Ag atoms are shown as large and small spheres, respectively, with the exception of the left view of (**e**), in which all atoms are shown as large spheres. In (**b**), the perfect $L1_1$@Ag-skin phase is shown. The numbers give the DFT energy differences from the lowest-energy homotops, which are the perfect $L1_1$@Ag-skin structure shown in (**b**) and the structure in (**c**). DFT density functional theory

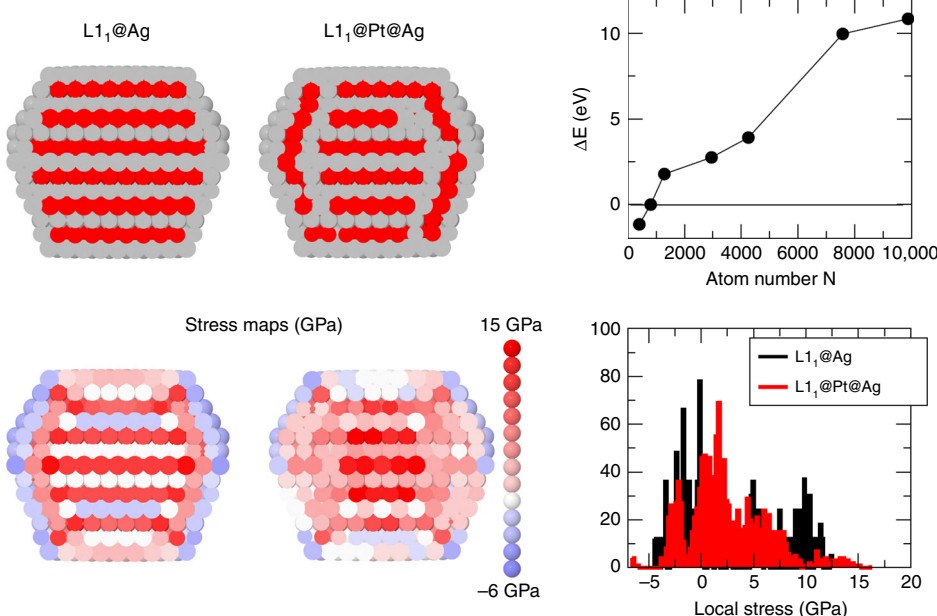

**Fig. 5** Theoretical predictions of $L1_1$ nanophases. $L1_1$@Ag core/shell and $L1_1$@Pt@Ag core/subshell/shell structures for the $TOh_{1289}$ (Pt atoms in red, Ag atoms in grey) and their associated stress map in colour code from −6 GPa for tensile stress to 15 GPa for compressive stress. The graph on the bottom right illustrates the stress distribution inside the $TOh_{1289}$ with the two structures and the graph on top shows the energy difference between the $L1_1$@Ag and the $L1_1$@Pt@Ag structures: positive sign means the $L1_1$@Pt@Ag is the stable structure

$L1_1$@Ag-skin phase is obtained for $Ag_{67}Pt_{12}$. The most significant structures for $Ag_{67}Pt_{12}$ are summarized in Fig. 4b–g. In all lowest-energy structures, all Pt atoms occupy subsurface positions. The best structures, very close in energy, are (b), (c) and (d): (b) is the perfect $L1_1$ phase with two Pt subsurface triangular leaflets, (c) and (d) are obtained from (b) by two and one atomic pair swaps, respectively. Structures with a Pt atom at the surface (e) or in the central site (f) are ~0.2 eV higher or more. A compact Pt aggregate (g) is ~1 eV higher than (b). These results show that the $L1_1$@Ag-

skin phase (either perfect or with a few defects) is energetically preferred for small clusters. $L1_1$@Ag-skin structures with a single twin plane are close in energy to perfect $L1_1$@Ag-skin structures (see Supplementary Fig. 3). On the contrary, our DFT global optimization runs show that $L1_1$ structures without Ag-skin (as in models 1 and 2 of Fig. 2) turn out to be energetically unfavourable (see Supplementary Fig. 4). Therefore, the DFT results very well agree with the experiments in showing that the Ag-skin is crucial for stabilizing the $L1_1$ phase.

To determine the driving force for the $L1_1$ phase breaking, we developed an atomistic model to calculate the pressure $p_i$ on each atom $i$[19]. Positive and negative $p_i$ correspond to compressive and tensile stress, respectively. The equilibration of $p_i$ is a driving force for optimizing nanoalloy chemical ordering[20]. According to the model, the $L1_1$@Ag-skin phase is energetically favourable for sizes below $10^3$ atoms, corresponding to $D \sim 3$ nm, in good agreement with the experiments. For larger sizes, more complex arrangements, as the $L1_1$@Pt@Ag-skin phase of Fig. 5, become more stable. In $L1_1$@Pt@Ag-skin, the subsurface layer is Pt-enriched, while the layer below is Ag-enriched. This is a kind of multidomain particle, with a central $L1_1$ core surrounded by thin $L1_1$ domains, with different orientations and three-layer thickness. To analyse stress equilibration, we use the average of the absolute value of $p_i$, $\overline{|p|} = \sum_{i=1}^{N} |p_i|$. Better stress equilibration corresponds to smaller $\overline{|p|}$[21]. For the TOh of 1289 atoms of Fig. 5, we obtain $\overline{|p|} = 4.07$ and 3.31 GPa for the $L1_1$@Ag-skin and the $L1_1$@Pt@Ag phases, respectively. Finally, we checked the compression of Ag−Ag and the expansion of Pt−Pt distances in the inner part of the $L1_1$@Ag-skin particles by DFT calculations on cuboctahedral and truncated octahedral clusters of 147 and 201 atoms, respectively. We found a contraction of Ag−Ag distances up to 2.5% and an expansion of Pt−Pt distances up to 1.5%. Contractions and expansions are measured with respect to nearest-neighbour distances in the corresponding pure Pt and pure Ag clusters.

In summary, the $L1_1$ phase is stabilized at its extreme nanoscale limit in AgPt nanoalloys. Surface segregation of excess Ag atoms creates an Ag-skin which is crucial for this stabilization. Otherwise, Pt atoms would be exposed on the surface, which is quite unfavourable due to the larger surface energy of Pt. From this point of view, AgPt is different from FePt and CoPt[4], in which the surface of $L1_0$-ordered nanoparticles is rather mixed or slightly Pt-enriched but without a pure skin segregation as in AgPt. As Ag atoms are larger than Pt atoms, the Ag-skin induces strong, size-dependent tensile/compressive stress in the alternating Pt/Ag (111) planes of the $L1_1$-ordered core, which becomes no longer energetically favourable with increasing size. This leads to the breaking of the $L1_1$ phase in multiple domains for $D > 3$ nm, a fact which also explains why this phase is very difficult to obtain in bulk samples.

## Methods

**Experiments**. AgPt nanoalloys were prepared by atomic condensation on an amorphous carbon surface using two separate atoms sources operating simultaneously under ultrahigh vacuum conditions as already described[22]. The amorphous substrate was kept at room temperature during all atomic depositions. The atomic composition was adjusted with the atomic flux of each source and checked by energy dispersive X-ray (EDS) and Rutherford Backscattering Spectroscopies and was kept at 55 ± 3 at.% Ag. After the preparation, some samples were annealed up to 400 °C to promote atomic diffusion. The annealing was performed through different steps: at 200 °C for 60 min, at 300 °C for 60 min and at 400 °C for 60 min. Each step was reached with a 2.5 °C/min ramp. TEM and GISAXS[4] were both performed to determine the NPs' size and shape. The substrates used for all characterization methods are amorphous carbon layers prepared in the same preparation batch. They present the same thickness and roughness. The only difference comes from that for GISAXS and RBS; the carbon layer is deposited on a flat silica substrate and for TEM and EDX the carbon layer is self-supported on grids. For fitting the GISAXS experiments, IsGISAXS software was used with a truncated sphere model as the NPs' shape[23]. High resolution in TEM mode (HRTEM) and High angle annular dark field in scanning transmission electron microscopy mode (STEM-HAADF) were performed, using a double aberration-corrected JEOL ARM200F microscope, to combine structural and chemical studies. STEM-HAADF images give a scattered intensity scale directly correlated to the atomic number of the elements composing the samples[24]. Since the HAADF contrast is both atomic number and thickness dependent, in order to quantitatively analyse these images, simulation of STEM-HAADF images were performed

using the QSTEM software[25]. QSTEM simulations were performed for different chemical arrangements such as ordered or disordered phases, in the alloyed or core-shell configurations.

**Density-functional theory (DFT) global optimization calculations**. The global optimization of chemical ordering was achieved by interfacing our basin hopping code[26] to the CP2K DFT code[27]. Some calculations (reported in Supplementary Note 2) have been checked also using the Quantum Espresso package[28]. The calculations were made by using the Perdew−Burke−Ernzerhof (PBE)[29] exchange-correlation functional. The basin hopping searches were made starting from a random chemical ordering and then allowing swaps between Pt−Ag atom pairs. For each composition, $10^3$ basin hopping steps were made.

**Atomistic modelling and Monte Carlo simulations**. To simulate larger clusters we used a semi-empirical potential in the second moment approximation (SMA)[30] in which we added a Gaussian term for the mixed second neighbour interactions to stabilize the $L1_1$ phase against the $L1_0$ phase. This potential has been fitted to DFT-PBE calculations[29] of the cohesive energies, lattice parameters and elastic constant of pure elements, and mixing enthalpies of the alloys for the mixed interactions. In order to reproduce the experimental order/disorder temperature of the $L1_1$ bulk phase (1200 K), we overestimate the formation enthalpy of the $L1_1$ phase ($-0.213$ eV per atom) as compared to the DFT-PBE calculation ($-0.051$ eV per atom). One of the $L1_0$ phase (0.063 eV per atom) is quite comparable to the DFT-PBE one which is equal to 0.041 eV per atom. The form of the potential is reported in Supplementary Note 3 and its parameters are given in Supplementary Table 2. Monte Carlo simulations were performed in the canonical ensemble. Elementary moves comprise both atomic exchanges of pairs of unlike atoms and atomic displacements. Atomic pressure was calculated as the trace of the atomic stress tensor[19].

## Data availability

All data are available from the authors upon reasonable request.

## Code availability

CP2K and Quantum Espresso codes are freely available at https://www.cp2k.org/ and https://www.quantum-espresso.org/, respectively. The BH and Monte Carlo codes are available upon request to ferrando@fisica.unige.it and mottet@cinam.univ-mrs.fr, respectively.

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

## Acknowledgements

We wish to express our gratitude to Dr. Eiji. Okunishi from Jeol Ltd., Tokyo, Japan for helpful discussions and access to the double corrected ARM200F microscope. The authors acknowledge financial support from the CNRS-CEA "METSA" French network (FR CNRS 3507) for the TEM experiments conducted on the MPQ-Paris Diderot platform. The authors would like to acknowledge support from the International Research Network—IRN "Nanoalloys" of CNRS.

## Author contributions

J.P., C.A.-V. and P.A. performed the experimental part, A.F. and C.M. developed the atomistic model, D.R. and R.F. made the DFT calculations. The authors wrote the paper together.

## Additional information

**Competing interests:** The authors declare no competing interests.

