## [Peer Review File · Nature Communications]

Reviewers' comments:

Reviewer #1 (Remarks to the Author):

This manuscript presents experimental and computational data to show the size-dependent stability of L1 phase AgPt nanoparticles. A critical size was identified using high resolution transmission electron microscopy, together with density functional theory calculation and molecular dynamic simulation. Overall, this work is self-consistent; the main conclusions are supported by the experimental data. I recommend the publication of this manuscript after minor revisions.

One question I have is related to the L1 phase breaking. While Ag skin was attributed to the enhanced stability and properly addressed by the simulation data as a key contributing factor, can the formation of twinning also be quantitatively analyzed computationally and discussed further? What are the differences between Ag skin and twinning formation in their contribution to the stability of L1 phase of AgPt? Other minor suggested changes include: In Figure 1, b) is an inset of the panel a) and its scale bar is missing. In Figure 2, panel b) should be presented before, not after, d) and c).

Reviewer #2 (Remarks to the Author):

In this work the authors have carried out a joint experimental and theoretical study of the size dependence of structural order in AgPt nanoclusters. The novelty of their result is that contrary to expectations, smaller clusters exhibit a well ordered intermetallic phase while larger clusters (with diameter > 2.5 nm) possess less ordered multi-domains. Conventional wisdom would say that as the size of the cluster increases, it would become more bulk like and ordered. One would then expect the ordered L11 phase to be found in the larger cluster. Instead the authors find this to be the case for the smaller nanoalloys. The results are compelling as they combine a number of state-of-the-art experimental imaging (STEM-HAADF and HRTEM) and structure (GISAXS) measurement techniques with multiscale computational approaches utilizing density functional theory (DFT) for accurate microscopic information and empirical potential for larger scale simulation of the site dependent stress in the system. The remarkable one-to-one correspondence between the measured and the calculated images showing the L11 phase is quite striking. For large scale simulations the authors develop interatomic potentials, with input from DFT, which they find to give results close to what they obtain from DFT. These simulations lead them to the conclusion that the L11 phase is stabilized in smaller AgPt nanoalloy via surface segregation of excess Ag atoms which creates an Ag skin, and a size dependent tensile/compressive stress in the alternating Pt/Ag (111) planes of the L11-ordered core. This balance of surface stress disappears for the larger nanoparticles and multiple domains are created to release the stress.

The above results are distinct from what has been found for other nanoalloys, for example, Fe-Pt and Co-Pt. In this regard, the manuscript would have been even more powerful had the authors traced the size dependent tensile/compressive stress to features in the electronic structure of the system, perhaps by carrying our DFT simulations for a few larger nanoparticles. Of course, these simulations are demanding. I am recommending publication as the results are novel and show great synergy between experiments and modeling in extracting structural information about nanoalloys of significant technological importance. The interaction potentials that the authors develop should also find much application in the field. The experimental data by themselves are also very convincing.

Reviewer #3 (Remarks to the Author):

The authors report the results of experimental and simulation work on AgPt nanoalloys forming L11 phase. Unlike other Pt nanoalloys which display L10 – ordered core, the L11 morphology in AgPt NPS is stabilized by an excess of Ag atoms segregated into a surface skin layer.

Due to the interest in catalytical properties of Pt alloys, the question regarding the existence of L11 (and other phases) in the middle-portion of AgPt phase diagram and its stability has been recurring over at least the two past decades of research. The authors have cited the publications: by Durussel and P. Feschotte (1996) and by Okamoto (1997) which reported the existence of only one metallic compound Ag₁₅Pt₁₇ in that region; this problem has recently been revisited by Wolverton et al. who reported experimental observation of L11 phase in bulk alloys at 50%Pt surrounded by disordered regions.

Here, the authors provide clear TEM, STEM and calculations evidence for the existence of L11 phase in NPs formed on carbon-coated TEM grids by RT vacuum deposition from two metal sources at near 50-50 atomic composition. They show that the phase is stable in particles smaller than certain size threshold before critical cluster diameter is reached and internal stresses prevail disrupting the ordering.

Overall, the findings in the article are well-supported by the experimental results with high-quality STEM and other data. I am leaving the judgment of the impact and broader interest of this work to the readership to the editor's discretion. I have the following questions regarding the scientific and technical merits of this work:

1) Have the authors considered the influence of support (e.g., a change to SiN_x membrane) on the nucleation of NPS and formation of this particular phase? It is clear that such interactions cannot be neglected at least in terms of their influence on particles' geometry i.e., Line 47: "because supported NPs look like truncated spheres due to the support-metal interaction."

2) It is unclear whether the GISAX data were collected directly on TEM grids or other solid support of the amorphous carbon film.

3) Lines 81-90, I agree that "The approximation of random orientations is crude" especially for NPs grown directly on the surface. Manual counting is always somewhat biased and as authors acknowledge, "We note that observing the L11 phase is possible only when the NP is oriented with a zone axis belonging to (111) planes". I was wondering why they didn't use tilting of the holder to align the zone axes of NPs in a statistically representative region of the grid?

Minor corrections:

Line 160 "to" is missing in "we wish express"

Line 172, "reach", should be "reached"

Line 173, Grazing Incidence X-ray scattering at small angles, non-conventional name of this technique, followed by the GISAXS acronym

Response to the Referees' comments:

We thank all Referees for their positive comments. Here below, our replies to the comments are in red, added/modified text of the article is in blue.

Referee #1 (Remarks to the Author):

This manuscript presents experimental and computational data to show the size-dependent stability of L1 phase AgPt nanoparticles. A critical size was identified using high resolution transmission electron microscopy, together with density functional theory calculation and molecular dynamic simulation. Overall, this work is self-consistent; the main conclusions are supported by the experimental data. I recommend the publication of this manuscript after minor revisions.

One question I have is related to the L1 phase breaking. While Ag skin was attributed to the enhanced stability and properly addressed by the simulation data as a key contributing factor, can the formation of twinning also be quantitatively analyzed computationally and discussed further? What are the differences between Ag skin and twinning formation in their contribution to the stability of L1 phase of AgPt?

Reply:

According to our calculations (both DFT and atomistic), the presence of Pt atoms in the surface layer is highly unfavourable, so that an Ag skin (or an Ag thicker shell, depending on composition) is formed in all cases. Therefore, also twinned nanoparticles have Ag at the surface. We added a new figure in the Supplementary Material (now Supplementary Figure 3) in which we compare by DFT, for size 79, the skin structures without twin plane ($L1_1@Ag$) and with twin plane ($L1_1^t@Ag$). In this case, $L1_1^t@Ag$ is obtained from $L1_1@Ag$ by two atomic pair swaps, and it is higher in energy by 0.04 eV. On the other hand, a single atomic swap from $L1_1@Ag$ to bring one Pt atom at the cluster surface costs at least 0.24 eV.

Therefore the energy scale of twinning is much smaller than the energy scale of surface segregation. The atomistic calculations for larger sizes confirm that, within the structures with Ag skin, perfect $L1_1$ domains and $L1_1$ domains with twin planes are rather close in energy for sizes below 3 nm, so that observing both structures is not surprising. The following sentence about twinned structures has been added also to the main article at lines 145-146

“ $L1_1 @Ag$ -skin structures with a single twin plane are close in energy to perfect $L1_1@Ag$ -skin structures (see Supplementary Figure 3).”

Other minor suggested changes include: In Figure 1, b) is an inset of the panel a) and its scale bar is missing. In Figure 2, panel b) should be presented before, not after, d) and c).

Reply:

Figure 1: In the revised version, the scale of b) is explained in the caption of figure 1

Figure 2: we corrected the figure 2 and provided a new figure.

Referee #2 (Remarks to the Author):

In this work the authors have carried out a joint experimental and theoretical study of the size dependence of structural order in AgPt nanoclusters. The novelty of their result is that contrary to expectations, smaller clusters exhibit a well ordered intermetallic phase while larger clusters (with diameter > 2.5 nm) possess less ordered multi-domains. Conventional wisdom would say that as the size of the cluster increases, it would become more bulk like

and ordered. One would then expect the ordered L11 phase to be found in the larger cluster. Instead the authors find this to be the case for the smaller nanoalloys. The results are compelling as they combine a number of state-of-the-art experimental imaging (STEM-HAADF and HRTEM) and structure (GISAXS) measurement techniques with multiscale computational approaches utilizing density functional theory (DFT) for accurate microscopic information and empirical potential for larger scale simulation of the site dependent stress in the system. The remarkable one-to-one correspondence between the measured and the calculated images showing the L11 phase is quite striking. For large scale simulations the authors develop interatomic potentials, with input from DFT, which they find to give results close to what they obtain from DFT. These simulations lead them to the conclusion that the L11 phase is stabilized in smaller AgPt nanoalloy via surface segregation of excess Ag atoms which creates an Ag skin, and a size dependent tensile/compressive stress in the alternating Pt/Ag (111) planes of the L11-ordered core. This balance of surface stress disappears for the larger nanoparticles and multiple domains are created to release the stress.

The above results are distinct from what has been found for other nanoalloys, for example, Fe-Pt and Co-Pt. In this regard, the manuscript would have been even more powerful had the authors traced the size dependent tensile/compressive stress to features in the electronic structure of the system, perhaps by carrying our DFT simulations for a few larger nanoparticles. Of course, these simulations are demanding. I am recommending publication as the results are novel and show great synergy between experiments and modeling in extracting structural information about nanoalloys of significant technological importance. The interaction potentials that the authors develop should also find much application in the field. The experimental data by themselves are also very convincing.

Reply: we have run a few DFT calculations for L1₁@Ag, pure Ag and pure Pt clusters of larger sizes (cuboctahedron of size 147 and truncated octahedron of size 201), in order to evaluate the strain of these structures, in particular to check the compression of Ag-Ag nearest-neighbour distances and the expansion of Pt-Pt nearest-neighbour distances. It turns out that for inner atoms there is a contraction of Ag-Ag distances up to 2.5% and an expansion of Pt-Pt distances up to 1.5%. These are measured with respect to nearest-neighbour distances in the corresponding pure Pt and pure Ag clusters. These results are now mentioned in the text at lines 162-167

“Finally, we checked the compression of Ag-Ag and the expansion of Pt-Pt distances in the inner part of the L11 @Ag-skin particles by DFT calculations on cuboctahedral and truncated octahedral clusters of 147 and 201 atoms, respectively. We found a contraction of Ag-Ag distances up to 2.5% and an expansion of Pt-Pt distances up to 1.5%. Contractions and expansions are measured with respect to nearest-neighbour distances in the corresponding pure Pt and pure Ag clusters.”

The DFT study of the effects of stress on electronic properties, and on chemical reactivity is indeed quite demanding, and it is beyond the scope of this work.

Referee #3 (Remarks to the Author):

The authors report the results of experimental and simulation work on AgPt nanoalloys forming L11 phase. Unlike other Pt nanoalloys which display L10 – ordered core, the L11 morphology in AgPt NPS is stabilized by an excess of Ag atoms segregated into a surface skin layer.

Due to the interest in catalytical properties of Pt alloys, the question regarding the existence of L11 (and other phases) in the middle-portion of AgPt phase diagram and its stability has been recurring over at least the two past decades of research. The authors have cited the publications: by Durussel and P. Feschotte (1996) and by Okamoto (1997) which reported the existence of only one metallic compound Ag₁₅Pt₁₇ in that region; this problem has recently been revisited by Wolverton et al. who reported experimental observation of L11 phase in bulk alloys at 50%Pt surrounded by disordered regions.

Here, the authors provide clear TEM, STEM and calculations evidence for the existence of L11 phase in NPs formed on carbon-coated TEM grids by RT vacuum deposition from two metal sources at near 50-50 atomic composition. They show that the phase is stable in particles smaller than certain size threshold before critical cluster diameter is reached and internal stresses prevail disrupting the ordering.

Overall, the findings in the article are well-supported by the experimental results with high-quality STEM and other data. I am leaving the judgment of the impact and broader interest of this work to the readership to the editor's discretion. I have the following questions regarding the scientific and technical merits of this work:

1) Have the authors considered the influence of support (e.g., a change to SiN_x membrane) on the nucleation of NPS and formation of this particular phase? It is clear that such interactions cannot be neglected at least in terms of their influence on particles' geometry i.e., Line 47: "because supported NPs look like truncated spheres due to the support-metal interaction."

Reply:

It is true that the support has some effect, but it should be a weak effect, since the NP shapes are not perfectly spherical but quite close to spheres anyway, as indeed written in the paragraph about annealed structures.

In particular, we note that the shape of the nanoparticles becomes somewhat more spherical after annealing. This indicates that the interaction with the substrate becomes weaker after annealing, probably due to the segregation of an Ag skin at the interface with the substrate. Since it is annealing which brings the appearance of the L1₁@Ag structures, this observation implies that the support has indeed little effect on the formation of the ordered phase.

In general, by looking at the literature, one can find several examples indicating that the amorphous carbon support has indeed a weak effect on metal nanoparticle structures. For example, in a recent paper (D. Foster et al., Nature Communications 9, 1323 (2018)), Au clusters with diameters of ~2.5 nm were deposited both on amorphous carbon and on silicon nitride, finding that the change of support has very little effect.

To better clarify this point, the paper is modified at lines 78-80, which now read

"... coalescence, leading to reorganization towards a morphology close to the quasi-spherical equilibrium shape of nanoparticles in vacuum^{4, 18}. The increase of H/D is indicative of some weakening of the nanoparticle/substrate interaction upon annealing."

2) It is unclear whether the GISAX data were collected directly on TEM grids or other solid support of the amorphous carbon film.

Reply:

The GISAXS measurements and TEM observations were performed on different samples, but obtained in the same preparation batch so under identical conditions: same substrate (amorphous carbon layer of same thickness and roughness checked by AFM measurements), same amount of deposited atoms (Pt and Ag), same deposition conditions (atom fluxes, pressure, and temperature), same annealing conditions (time, temperature)...

The only difference comes from that for GISAXS, the carbon layer is deposited on a flat silica substrate and for TEM the carbon layer is self-supported on grids.

To improve clarity, we specify in “methods/ experiments” item that the amorphous carbon substrate is identical for both, GISAXS and TEM samples. This sentence is added:

“ The substrates used for all characterization methods are amorphous carbon layer prepared in the same preparation batch. They present same thickness and roughness. The only difference comes from that for GISAXS and RBS, the carbon layer is deposited on a flat silica substrate and for TEM and EDX the carbon layer is self-supported on grids”

3) Lines 81-90, I agree that “The approximation of random orientations is crude” especially for NPs grown directly on the surface. Manual counting is always somewhat biased and as authors acknowledge, “We note that observing the L11 phase is possible only when the NP is oriented with a zone axis belonging to (111) planes”. I was wondering why they didn’t use tilting of the holder to align the zone axes of NPs in a statistically representative region of the grid?

Reply:

We checked whether the tilt variation could change the NPs’ proportion oriented with a zone axis allowing the clear evidence of the L1₁ phase: there is no quantitative change. The particles are to a good approximation randomly oriented on the carbon layer and the electron diffraction (done on a selected area of several tens of microns in diameter) confirms this point. So we have considered that the manual counting is independent of the tilt and can be representative and quantitative. This is why the counting has been done at zero tilt.

Minor corrections:

Line 160 “to” is missing in “we wish express”

Line 172, “reach”, should be “reached”

Line 173, Grazing Incidence X-ray scattering at small angles, non-conventional name of this technique, followed by the GISAXS acronym

Reply: these have been corrected

REVIEWERS' COMMENTS:

Reviewer #2 (Remarks to the Author):

I believe the authors have addressed the questions that were raised. I recommend that the article be accepted for publication.

Reviewer #3 (Remarks to the Author):

The authors have addressed all comments and remarks that I listed.

REVIEWERS' COMMENTS:

Reviewer #2 (Remarks to the Author):

I believe the authors have addressed the questions that were raised. I recommend that the article be accepted for publication.

Reviewer #3 (Remarks to the Author):

The authors have addressed all comments and remarks that I listed.

REPLY

We thank the reviewers for their positive comments. There are no changes to be made.